Research on license plate recognition based on graphically supervised signal-assisted training

Chi Dianwei 1 dianwei.chi@163.com
Jia Zehao 1 19863817078@163.com
Liu Lizhen 2
1 Artificial Intelligence Institute, Yantai Institute of Technology , Yantai, Shandong , China
2 Institute of Microbiology, Jiangxi Academy of Sciences , Nanchang, Jiangxi , China
Bhattacharyya Siddhartha
Electronic publication date: 2025 Jul 1
Publication date: 2025
Volume: 11
Electronic Location ID: e2989
Received 2024 Nov 22; Accepted 2025 Jun 6
Copyright: © 2025 Chi et al.
Copyright year: 2025
Copyright holder: Chi et al.
License: This is an open access article distributed under the terms of the Creative Commons Attribution License, which permits unrestricted use, distribution, reproduction and adaptation in any medium and for any purpose provided that it is properly attributed. For attribution, the original author(s), title, publication source (PeerJ Computer Science) and either DOI or URL of the article must be cited.
License URL: https://creativecommons.org/licenses/by/4.0/

Keywords: Deep learning, Supervised signaling, Assisted training, License plate recognition

Funding: Jiangxi Province 2023YSBG10007 This work was supported by the Foundation of Jiangxi Province (2023YSBG10007). The funders had no role in study design, data collection and analysis, decision to publish, or preparation of the manuscript.

==============================
Background

With the rapid growth of the number of cars and the increasing complexity of urban transportation, it is particularly important to achieve high-accuracy license plate recognition in complex scenarios. However, since license plate recognition models are mostly deployed on embedded devices with limited computational resources, designing a lightweight and accurate model has become an urgent problem in the field of license plate recognition.

Methods

This study proposes an improved license plate recognition algorithm. We use the License Plate Recognition Network (LPRNet) as the base model. To enhance its accuracy, we incorporate graphically supervised signals for assisted training. This approach refines the training process, yielding a model that is both lightweight and highly accurate. An auxiliary training branch is added, utilizing these graphical signals to guide the model in learning improved image features.

Results

Experiments show that compared with LPRNet, this study improves the accuracy in all test sets of the Chinese City Parking Dataset (CCPD) dataset, where the average accuracy is improved by 5.86%, the maximum accuracy by 10.9%, the average character precision by 2.1%, and the average recall by 6.9%, indicating that this study can achieve higher accuracy while keeping it lightweight. This study also provides new ideas for other deep learning image recognition tasks.

Introduction

With the rapid growth of the number of cars and the increasing complexity of urban transportation, vehicle management and traffic monitoring are particularly important. Thanks to the development of the Internet and deep learning technology, smart transportation has become a major trend in vehicle management and traffic monitoring (Wu, Wu & Wang, 2019; Yang, Xie & Xu, 2024). As a key link in the intelligent transportation system, license plate recognition technology is of great significance for the realization of vehicle violation monitoring, parking lot management, traffic congestion analysis, etc.

Deep learning methods are highly regarded for their powerful feature learning abilities and efficient modeling of complex data. With deep learning models, the system can automatically learn the character features on the license plate from a large amount of data and can accurately recognize them under different lighting, angles, and even partial occlusion. Compared with traditional methods, deep learning methods show higher robustness and accuracy in the field of license plate recognition and can effectively cope with the challenges in complex environments. At the same time, most of the application scenarios for license plate recognition are embedded devices with limited computational resources, which requires the deep learning model to be lightweight while improving performance.

For research on deep learning license plate recognition, the traditional approach to license plate recognition is usually performed in a two-stage process: first character segmentation of the license plate, followed by recognition of individual characters. For example, Zhang et al. (2018) used a morphological algorithm to locate the license plate region and segmented the license plate characters using a vertical projection-based method. Khan et al. (2018) combined the features of the image to segment the characters and used a support vector machine to classify the characters. Raza et al. (2020) used color space with a vertical projection method for license plate recognition and an improved deep learning model for single character recognition, which improved the accuracy, but the recognition results were still limited by the character segmentation results. This way of doing character segmentation first and then single-character recognition not only has low accuracy but also low detection efficiency.

Subsequently, a series of tasks for license plate recognition via neural networks emerged. Shi, Bai & Yao (2016) proposed the convolutional recurrent neural network (CRNN) model, which solves the problem of end-to-end license plate recognition by considering the problem such as license plate recognition as a multi-input and multi-output image sequence recognition problem. Zherzdev & Gruzdev (2018) proposed the LPRNet model, which used purely convolutional operations for the first time to recognize license plates. More recently, Vargoorani & Suen (2024) explored approaches combining detection models like Faster R-CNN with recognition models such as convolutional neural network-recurrent neural network (CNN-RNN) architectures leveraging Connectionist Temporal Classification (CTC) loss.

In recent years, in order to improve the adaptability of the recognition model for complex scenes, some researchers have proposed a license plate recognition model that incorporates denoising structures. For example, the license plate recognition system designed by Rao et al. (2021) achieved better license plate detection and recognition accuracy by improving the CNN detection network, STN, and bidirectional recurrent neural network (BRNN). Pirgazi, Pourhashem Kallehbasti & Ghanbari Sorkhi (2022) combined convolutional neural network with LSTM to achieve end-to-end license plate recognition but with low computational efficiency. Zhang et al. (2021) changed the standard CNN in CRNN to a micro-modified model of a depth-separable convolutional network and used a bidirectional long- and short-term memory network for RNN to improve the recognition accuracy of the model. Li et al. (2019) proposed an enhanced convolutional neural network model, AlexNet-L, which is an end-to-end network model for license plate character recognition, to improve the accuracy of license plate recognition. Wang & Chen (2024) proposed a special license plate recognition algorithm that uses deep residual networks and attention mechanisms to optimize the model structure and improve recognition accuracy. Kim, Kim & Park (2024) proposed a new license plate recognition method, AFA-Net, which achieves clarity restoration and significantly improves the recognition accuracy of feature attention networks for low-resolution and motion blurred images. Xu et al. (2024) proposed a CRNN-based method for ship license plate image recognition, improved the CRNN model through image enhancement and data enhancement, and achieved an accuracy of 93% on a wide range of image datasets. Lee et al. (2019) proposed an algorithm called SNIDER for license plate recognition from low-quality images, which integrates denoising and rectification tasks through a training network framework, effectively improving the model’s performance. Zhang, Liu & Ma (2018) combined license plate super-resolution and recognition in an end-to-end framework to improve the accuracy and adaptability of the model. Min, Lim & Gwak (2020) proposed the Fast-SRGAN model, which uses super-resolution to remove the noise present in the image and improve the final performance. Similarly, to tackle low-resolution and blurry images, Nascimento et al. (2024) introduced a layout-aware and character-driven approach for license plate super-resolution, enhancing character reconstruction. Xu et al. (2021) proposed a 3D perspective transform-based License Plate Correction Algorithm (APAN), which improves recognition accuracy through a GRU structure based on a 2D attention mechanism. Ceng & Ke (2021) introduced Vision Transformer into the study of license plate recognition, which suppresses noise and improves accuracy through self-attention mechanisms. Furthermore, AlDahoul et al. (2024) highlighted the exploration of vision language models (VLMs) as a promising direction for recognizing unclear plates, especially under challenging conditions. In addition, some studies improve the model’s adaptability in complex situations through data augmentation. For example, Yang, Wang & Jiang (2024) utilized a haze enhancement algorithm to improve the recognition accuracy of the model in haze scenarios. Zhang et al. (2020) proposed a CycleGAN model for generating license plate images and a license plate recognizer based on an Xception-based CNN encoder with a 2D attention mechanism, which experimentally proved to have good performance in a variety of environments. Jin et al. (2021) applied a dark channel a priori algorithm to preliminary dehaze fuzzy images and realized image super-resolution by a super-resolution convolutional neural network to improve the accuracy in hazy scenes. Zhu, Wang & Wang (2024) improved the YOLOv5 algorithm to improve the accuracy of license plate localization but did not recognize license plate characters.

In summary, current deep learning models have achieved significant performance improvements on license plate recognition tasks, but these improvements often come at the cost of increasing the number of parameters and computational complexity of the model. This approach not only increases the storage and running cost of the model, but also becomes impractical to deploy on resource-constrained embedded devices. Although some studies have proposed lighter model structure designs, such as LPRNet and CRNN, which are able to be deployed in embedded devices, the insufficient learning ability and generalization of the overly lightweight models result in lower accuracy rates.

Therefore, how to further improve the accuracy of models while keeping them lightweight has become an urgent problem in the field of license plate recognition. This study focuses on optimizing the training process of existing lightweight models to improve their recognition accuracy and generalization ability.

In this article, we propose a research on license plate recognition based on graphical supervised signal assisted training, by introducing the labeling of image modality and auxiliary training branch to assist the training of the model, the auxiliary training branch only plays a role in the training phase, and the relevant structure of the auxiliary training branch will be removed during the testing phase, in order to ensure that the model is lightweight. This research effectively improves the accuracy of the lightweight model.

The main contributions of this article are: Novel application of graphical supervision: We explore and utilize graphical supervisory signals derived directly from text labels (via fonts) to assist LPR model training. This leverages readily available label information in a new modality.

Accuracy enhancement without inference cost: We demonstrate that incorporating this graphical supervision via an auxiliary training branch significantly improves LPR accuracy on the challenging CCPD dataset, particularly in complex scenarios, without increasing the parameter count or computational cost (FLOPs) of the final inference model compared to the baseline LPRNet.

Proposed training framework: We present a dual-branch training framework that effectively integrates standard CTC-based sequence learning with auxiliary graphical feature learning, showing its practical effectiveness for enhancing lightweight sequence recognition models. This provides a potential methodology for improving other deep learning vision tasks where label information can be represented graphically.

The remainder of the article is organized as follows. “Related Work” briefly introduces the related work. “Methods” describes the design flow of the methodology of this article. “Experiments and Discussions” evaluates the methodology of this article through experiments. “Discussion” concludes the article and outlines future research directions. In addition, the exact meaning of the Chinese characters in the article and the code is given in the Supplemental Material.

Related work

Mainstream LPR architectures

Early deep learning approaches often adopted end-to-end recognition models. Seminal works like CRNN (Shi, Bai & Yao, 2016) combined CNNs for feature extraction with RNNs for sequence modeling, typically trained using CTC Loss (Graves et al., 2006) to handle variable-length character sequences without explicit segmentation. Addressing the computational overhead of RNNs, LPRNet (Zherzdev & Gruzdev, 2018) introduced a purely convolutional architecture for efficient, real-time LPR, also leveraging CTC Loss. Other approaches follow a two-stage pipeline, first detecting the license plate region and then recognizing the characters within it, sometimes employing models like Faster R-CNN for detection followed by specialized recognition networks (e.g., Vargoorani & Suen, 2024).

Strategies for improving LPR performance in challenging conditions

A primary thrust in LPR research is enhancing robustness against real-world image degradations.

Image restoration and super-resolution: Techniques aimed at improving input image quality are common. For instance, Nascimento et al. (2024) focused on layout-aware super-resolution, while Kim, Kim & Park (2024) developed AFA-Net for joint deblurring and super-resolution. Generative adversarial networks (GANs) have also been employed for super-resolution tasks in LPR (Zhang, Liu & Ma, 2018; Min, Lim & Gwak, 2020). Lee et al. (2019) integrated denoising and rectification directly within their SNIDER framework. These methods typically add preprocessing steps or incorporate complex restoration modules.

Attention mechanisms and architectural enhancements: Attention mechanisms have been widely adopted to help models focus on salient character regions (Wang & Chen, 2024; Zhang et al., 2020; Xu et al., 2021). Some works also explore novel architectures, such as the use of Vision Transformers (Ceng & Ke, 2021) or, more recently, visual language models like VehiclePaliGemma (AlDahoul et al., 2024), for tackling difficult recognition tasks.

Our approach differs from these works. Instead of modifying the input image or significantly altering the inference architecture of a lightweight model like LPRNet, we focus on enriching its learned representations through an auxiliary training signal derived from existing labels.

Leveraging auxiliary information and training paradigms

The concept of using additional information or tasks to aid model training is well-established. Font characteristics have been identified as an influential factor in LPR performance (Vargoorani & Suen, 2024).

In broader computer vision, self-supervised learning (SSL) methods excel at learning representations from unlabeled data by designing pretext tasks (e.g., He et al., 2020; Chen et al., 2020; He et al., 2022). While SSL typically derives supervision from the input data itself, our work is inspired by the principle of mining richer supervisory signals.

Our proposed graphically supervised signals (GSS) can be viewed within the context of auxiliary learning. We introduce a secondary task—reconstructing an image representation of the text label—to run in parallel with the primary LPR task. The key distinction of our GSS approach is the direct transformation of textual labels into a graphical (image) modality to provide explicit, pixel-level guidance. This contrasts with methods that might use abstract intermediate representations or different types of auxiliary losses not directly tied to the visual rendering of the target text.

Focus on lightweight model enhancement

The practical demand for efficient LPR systems necessitates a focus on lightweight models (Zherzdev & Gruzdev, 2018; Zhu, Wang & Wang, 2024). Many performance-enhancing techniques, however, lead to increased model complexity. Our research specifically addresses the challenge of improving the accuracy of an existing, proven lightweight architecture (LPRNet) by optimizing its training process with GSS, aiming to achieve a better accuracy-efficiency balance without modifying the inference pipeline.

We selected LPRNet as the base framework for this study due to several key advantages. Firstly, it achieves excellent performance on benchmark LPR datasets while being remarkably lightweight, requiring only approximately 0.34 GFLOPs for a single forward pass (Zherzdev & Gruzdev, 2018). This efficiency makes it highly suitable for deployment on embedded devices with limited computational resources, striking a crucial balance between high recognition accuracy and low resource consumption. Secondly, LPRNet performs end-to-end recognition without requiring prior character segmentation, a common bottleneck in traditional LPR systems. It is characterized by its high accuracy, real-time performance, and inherent support for variable-length license plates. Furthermore, LPRNet has demonstrated its capability for end-to-end training on license plates from different countries, accommodating significant variations in character appearance (Zherzdev & Gruzdev, 2018). As one of the pioneering real-time lightweight LPR algorithms that does not rely on RNNs, its purely convolutional architecture allows it to run efficiently on diverse hardware platforms.

LPRNet has been widely adopted and validated within the LPR research community. Numerous subsequent studies have employed LPRNet either as a baseline for comparison or as a foundational component for further improvements specifically within the domain of license plate recognition. Its application has been predominantly focused on LPR tasks rather than general object recognition, owing to its specialized architecture optimized for recognizing linear sequences of characters typical of license plates. This established track record and its specific design for LPR make it an ideal choice for our work aiming to enhance training methodologies for efficient and accurate license plate recognition.

The backbone network of LPRNet, depicted in Fig. 1, processes an original RGB image to extract discriminative features for license plate recognition. This network is primarily a CNN architecture.

Figure 1 LPRNet network structure diagram.

As shown on the left side of Fig. 1, the input image sequentially passes through several key modules: CBR blocks: These consist of a 2D Convolutional layer (Conv2d), followed by Batch Normalization (BatchNorm2d) and a rectified linear unit (ReLU) activation function, as detailed on the upper right of Fig. 1.

MaxPool3d layers: These perform 3D max pooling to reduce spatial dimensions.

Small basic blocks: These are custom residual-like blocks, detailed in the middle right of Fig. 1. Each block comprises a sequence of Conv2d layers, interspersed with ReLU activations, and concluding with BatchNorm2d and another ReLU. This structure helps in learning more complex features.

DropOut layers: These are used for regularization to prevent overfitting.

A distinctive feature of LPRNet is its replacement of traditional RNNs for sequence modeling. Instead, LPRNet utilizes a specialized 1 × 13 convolutional kernel (not explicitly shown as a single block in this high-level diagram but is a core part of the Conv2d layers within the architecture, particularly in later stages or the classifier) to capture contextual information across the width of the feature map, effectively acting as a context-bound operator.

Furthermore, LPRNet incorporates global context embedding. As illustrated in the lower right part of Fig. 1, features from different stages or parallel paths (after passing through Average Pool layers) are processed by scaler modules. The scaler module, detailed on the right, normalizes the features using a power function (Power(2)), mean calculation, and division (div), which can be interpreted as a form of local response normalization or feature scaling. These scaled features are then concatenated (Concat) before being fed into the final classifier. This concatenation of globally pooled and scaled features enhances the network’s expressive power by providing a richer set of information to the classifier.

Finally, similar to many sequence recognition models, LPRNet is trained end-to-end using CTC Loss. This loss function is necessary because the length of the network’s encoded output (feature sequence from the backbone) is typically not equal to the variable length of the license plate character sequences. In the field of license plate recognition, the LPRNet model stands out with its relatively high accuracy and lightweight design, and has become one of the most widely used models in the field. Its efficient computational performance and low resource consumption have led to its successful deployment in many embedded devices. In view of this, LPRNet is chosen as the base model in this study, aiming to further improve its recognition accuracy through in-depth optimization of its training process.

Methods

Graphically supervised signals

Effectively utilizing supervised signals is a key research area, especially with the rise of big data. Self-supervised Learning has emerged as a prominent approach in recent years. In machine vision, SSL often focuses on extracting supervisory signals directly from input data. A notable example is the MoCo algorithm by He et al. (2022). This method uses contrastive learning and a dynamic dictionary for unsupervised representation learning. Chen et al. (2020) proposed the SimCLR algorithm, which improves the effect of contrast learning by data augmentation with the construction of larger batches. Chen & He (2021) further elaborated the self-supervised algorithm, which solves the problem of gradient collapse by using gradient stopping. He et al. (2022) again re-explored the supervised signals in the image by first randomly masking the image and then predicting the masked portion from the unmasked portion. In the field of natural language processing (Devlin et al., 2018; Radford et al., 2018), although not using the idea of self-supervised learning, it is similar to self-supervised learning, i.e., the model is first trained by a large-scale pre-training task, which allows the model to acquire a certain amount of learning ability in advance by pre-training on an unsupervised, large-scale textual dataset, before fine-tuning it on a specific task.

The license plate recognition task has some special characteristics. Firstly, the resolution of license plate images is small, and small resolution license plate images cannot adapt to most of the auxiliary tasks, which makes self-supervised learning difficult to realize in license plate images; secondly, the signal-to-noise ratio of license plate images in complex scenes is lower compared to that of conventional images, and conventional auxiliary tasks may make the model learn more noise, which in turn reduces the performance of the model.

The research in this article is inspired by self-supervised learning, hoping to mine some additional valuable information in the existing dataset and use this additional information to assist the training of the model, in order to improve the model’s performance at the time of testing. However, the research in this article is not exactly the same as self-supervised learning, and the difference between the research in this article and self-supervised learning is specifically manifested in the following two aspects: The supervisory information of self-supervised learning comes from data, while the supervisory information of this study comes from labels.

Self-supervised learning mostly uses the training paradigm of unsupervised pre-training-supervised fine-tuning, while this study merges the pre-training process with fine-tuning, and performs pre-training and fine-tuning at the same time in the form of dual training branches to ensure the effectiveness of the model.

In our pursuit of enhanced performance in license plate recognition (LPR), this study explores the mining of novel supervisory signals from existing information. Conventional deep learning-based LPR, exemplified by models like LPRNet, typically relies on standard supervised learning. In such setups, red green blue (RGB) images serve as input, producing fixed-length feature sequences that are then matched with variable-length text sequence labels through a CTC Loss. The network parameters are subsequently updated via backpropagation, driven entirely by these text-based, semantic-level supervisory signals. While effective, this semantic supervision primarily guides the model towards understanding the textual content.

To further assist the model in learning robust semantic features and thereby improve both recognition accuracy and overall system resilience, we introduce an auxiliary learning mechanism. This mechanism is driven by what we term “graphically supervised signals” (GSS). The core idea is to extract and utilize the inherent graphical information contained within the textual labels themselves to augment the learning process.

For LPR and similar text recognition tasks, text labels inherently possess visual characteristics, primarily defined by their font. By leveraging font rendering, we can transform these textual signals into specific graphical representations—effectively constructing an image from the text. This “image label” then serves as the GSS. These GSS provide a direct, pixel-level target for an auxiliary decoder branch, compelling the shared feature extractor within LPRNet to learn the fine-grained visual characteristics and spatial layout of the characters. This use of label-derived graphical images as explicit supervisory targets for an auxiliary task represents our specific strategy to improve feature learning in the LPR context, complementing the primary semantic supervision provided by the CTC Loss. The concept of GSS and its distinction from purely semantic signals are illustrated in Fig. 2.

Figure 2 Two representations of string labels.

The image label of license plate is constructed precisely through fonts, and graphical supervisory signals are introduced in an attempt to obtain supervisory signals in fonts with respect to the graphical features of the text. As shown in Fig. 3, in constructing the image label, we spatially aligns the real training image with the image label. The goal is to make the location of the license plate pixels in the image label as close as possible to the location of the license plate pixels in the original map.

Figure 3 Spatial alignment of image label.

The input to the spatial alignment algorithm is the average image of the training set, and the output is the alignment boundary. The training set is first traversed to find the average image of all license plates, using the average map as an abstract representation of the entire training set. Subsequently, the average map is processed to remove irrelevant information from the image by grayscaling and binarization. Next, the image is projected, respectively, on the horizontal axis and the vertical axis of the two directions. The two projection curves, respectively, set the threshold values of the two directions, and two threshold straight lines can be obtained. Through the threshold straight line and the intersection of the projection curve, you can determine the spatial alignment of the boundary. Then determine the alignment boundary. You can generate the same size with the boundary of the characters, which will be placed in the image in accordance with the alignment boundary.

Auxiliary training

Upsampling decoder

How to effectively utilize graphically supervised signals is a central issue in the task flow. This article draws on the ideas in autoencoder (Michelucci, 2022). Autoencoder belong to unsupervised learning, which predicts the input data through the output data so that the output is as similar as possible to the input, and the purpose of autoencoder is generally to compress the input data by using the hidden layer. Autoencoder put the supervised signals they carry to good use by predicting the input. For the problem of how to utilize the graphical supervisory signal, this article refers to the idea of a autoencoder. The difference is that the autoencoder predicts the input, while this article predicts the image label.

In order to effectively utilize the image label as a supervised signal for training, we design an upsampling decoder in the feature output layer of the original neural network. The construction of the upsampling decoder is accomplished using only a simple three-layer transposed convolution and a projected convolution. The input to the upsampling decoder is the feature map output from the LPRNet backbone, which is used as the input to both the text sequence classifier and the upsampling decoder. The location of the upsampling decoder and its structure are shown in Fig. 4. After processing by the upsampling decoder, the feature map is sampled to the same size as the image label. Subsequently, the output of the upsampling decoder is made as close as possible to the image label by MSE Loss.

Figure 4 Position of the upsampling decoder and schematic of its structure.

MSE Loss in upsampling decoder can be expressed as:

(1) Lmse=1n×h×w∑i=1h⁡∑j=1w⁡(yi,j−y^i,j)2

where, n denotes the number of license plate samples, h and w respectively denotes the height and width of the image label, y denote the outputs of upsampling decoder, y denote image label. The closer the upsampling decoder output is to the image label, the smaller the loss.

Algorithmic process

The current conventional license plate recognition process is to input the RGB image of the license plate, get the output of the category sequence through the deep learning model, and then match the category sequence with the label sequence through CTC Loss and calculate the loss.

Suppose the input sequence of CTC is X=(x1,x2,…,xT), where T is the CTC input length. the input of CTC is the output of the license plate recognition model, which is a fixed length. Suppose the output sequence of CTC is Y=(y1,y2,…,yL), where L is the CTC output length and L≤T. The purpose of CTC is to find a mapping from X to Y. CTC generates alignment paths by 1. inserting a blank character ∅, at each position in the sequence of output characters to indicate that no character is output at a certain time step. 2. allowing the neighboring output characters to remain the same to indicate continuous output of characters. With these operations, CTC can align fixed-length input sequences with shorter and variable-length output sequences. The output of CTC is actually a probability distribution of a series of labeled sequences, each of which can generate different alignment paths by inserting blank characters and repeating characters. The final output is the probability sum of all possible paths.

The goal of CTC Loss is to maximize the total probability of an alignment path for a sequence Y of correct characters. Let B denote the function that maps alignment paths to output sequences and P(Z|X) denote the probability of generating alignment path Z given input X, The CTC Loss can be expressed as:

(2) P(Y|X)=∑Z∈B−1(Y)⁡P(Z|X)

where, B−1(Y) denotes the set of all alignment paths that can be mapped to the output sequence Y. To compute this probability, CTC computes the probability of each aligned path and sums it by means of dynamic programming.

After calculating the probabilities of all possible paths, the final form of the CTC Loss is the negative log-likelihood function:

(3) CTCLoss=−logP(Y|X).

The advantage of CTC Loss is that sequence alignment can be performed directly in the training phase without additional alignment labels. It can effectively deal with sequence alignment problems and can automatically learn the alignment corresponding to the target sequence during the training process. In license plate recognition, CTC Loss can be used to train neural network models to improve the accuracy of license plate recognition by optimizing CTC Loss. By introducing CTC Loss, the license plate recognition model can learn the position of the characters and the alignment relationship between the characters, thus improving recognition accuracy.

The characteristic in the algorithm flow in this article is that a new auxiliary training branch is designed, and the auxiliary training branch is used to receive graphic supervised signals. The training process of the model consists of the auxiliary training branch and the regular training branch, and the training process is shown in Fig. 5. The models in the regular training branch and the auxiliary training branch are slightly different, and this article calls the model in the regular training branch the regular model and the model in the auxiliary training branch the auxiliary model. The inputs of the regular model and the auxiliary model are exactly the same, and the backbone part of the model is weight-sharing, which means that the same backbone completes the two training tasks of the two training branches at the same time, i.e., the two training branches are training the same backbone. The difference between the regular model and the auxiliary model is that the regular model inputs the output of the backbone directly into the classifier to get the category sequence, while the auxiliary model inputs the input out of the backbone into the upsampling decoder, which decodes the feature map output from the backbone into an image of the same size as the image label to receive the graphical supervisory signals.

Figure 5 Flow of algorithm for graphical supervised signal-assisted training.

During model training, the two training branches alternate. That is, for a certain batch of data in the training set, the model weights are first updated through the regular training branch, and then using the same data, the model weights are updated through the auxiliary training branch. When both branches have finished updating the weights, the next batch of data is obtained, and the above process is repeated. In the testing phase, only the regular training branch is retained, and the auxiliary training branch is no longer needed. Therefore, no additional computation is added to the model in the testing phase.

To demonstrate that the auxiliary graphical supervision signal effectively contributes to the learning of the primary license plate recognition task, we analyze how gradients from both loss functions influence the parameters of the shared backbone network.

Let θB denote the parameters of the shared backbone network fB(x;θB), which takes an input image x and extracts features. These features are then utilized by two separate heads: (1) A classifier head fC (with parameters θC) for the primary license plate recognition task, optimized using the CTC loss, Lctc(θB,θC).

(2) An auxiliary decoder head fD (with parameters θD) for the graphical reconstruction task, optimized using the mean squared error (MSE) loss, Lmse(θB,θD).

Critically, the backbone parameters θB are shared between these two tasks. This means that θB appears in the computation graph of both Lctc and Lmse. Consequently, during backpropagation, gradients from both loss functions will be computed with respect to these shared parameters θB.

Our proposed training methodology, detailed in Algorithm 1, employs a sequential update strategy for the shared backbone parameters within each training iteration (batch). This process can be described as follows:

Algorithm 1 Graphically supervised assisted training.

Input: Dataset D={(x,yctc,ymse)}, Learning rate α	
Parameters: Backbone θB, Classifier θC, Decoder θD	
Model: z=fB(x;θB),p=fC(z;θC),y^=fD(z;θD)	
Losses: LCTC(p,yctc),LMSE(y^,ymse)	
Output: Optimized Inference Parameters θB∗,θC∗	
1: Initialize θB,θC,θD	
2: for each epoch do	
3: for each batch B=(xi,yctc,i,ymse,i)i=1N⊂D do	
4: // 1. Primary Task Update (CTC)	
5: Z←fB(XB;θB)	
6: P←fC(Z;θC)	
7: L1←Average(LCTC(P,Yctc,B))	
8: (θB,θC)←(θB,θC)−α∗∇(θB,θC)L1	
9: // 2. Auxiliary Task Update (MSE)	
10: Z′←fB(XB;θB) // Features using updated θB	
11: Y^←fD(Z′;θD)	
12: L2←Average(LMSE(Y^,Ymse,B))	
13: (θB,θD)←(θB,θD)−α∗∇(θB,θD)L2	
14: end for	
15: end for	
16: Return θB, θC	

Let θB(k) be the state of the backbone parameters at the beginning of a training step k for a given batch. 1. Update from primary task (CTC loss):

The backbone parameters are first updated based on the gradient from the CTC loss: (4) θB(k+1/2)=θB(k)−α∗∇θBLctc(θB(k),θC)

where α is the learning rate, and ∇θBLctc(θB(k),θC) is the gradient of Lctc with respect to θB evaluated at θB(k).

2. Update from auxiliary task (MSE loss):

Subsequently, the newly updated backbone parameters θB(k+1/2) are further refined based on the gradient from the MSE loss. The features for this step are computed using fB(x;θB(k+1/2)). (5) θB(k+1)=θB(k+1/2)−α∗∇θB′Lmse(θB(k+1/2),θD)

where ∇θB′Lmse(θB(k+1/2),θD) denotes the gradient of Lmse with respect to the backbone parameters, evaluated at their current state θB(k+1/2).

By substituting Eq. (4) into Eq. (5), we can express the net change to the backbone parameters after one full cycle of updates within the batch:

θB(k+1)=(θB(k)−α∗∇θBLctc(θB(k),θC))−α∗∇θB′Lmse(θB(k+1/2),θD)

(6) θB(k+1)=θB(k)−α∗(∇θBLctc(θB(k),θC)+∇θB′Lmse(θB(k+1/2),θD)).

Equation (6) clearly demonstrates that the total update applied to the shared backbone parameters θB in a single training step is effectively a combination of the gradients derived from both the primary task ( Lctc) and the auxiliary task ( Lmse). Although the MSE gradient is computed based on slightly updated parameters ( θB(k+1/2)), the crucial point is that both supervisory signals contribute distinct gradient information that shapes θB.

Therefore, the graphical supervision signal, through its Lmse gradient, actively participates in and influences the learning process of the shared feature extractor ( fB). This co-supervision encourages the backbone network to learn richer and more robust feature representations that are beneficial not only for reconstructing graphical elements but also for improving the accuracy of the primary license plate recognition task. The optimized backbone parameters θB∗ used during inference are thus a result of this synergistic training process.

The practical implementation of this dual-branch training, detailing how these updates are orchestrated, is formally described in Algorithm 1.

Experiments and discussions

Training dataset and evaluation dataset

In this article, we use the CCPD dataset for training, which is a large, diverse, and carefully labeled open-source dataset of Chinese urban license plates (Xu et al., 2018). Data is available at https://github.com/detectRecog/CCPD. After opening the link, find CCPD (ECCV), which can be downloaded directly through the Google Drive link. In order to conform to the license plate recognition problem studied in this article, the dataset is processed in this article. The target region of the license plate is cropped by the corner points of the license plate labeled by the CCPD dataset and corrected by an affine transformation.

Part of the license plate data is shown in Fig. 6. The CCPD dataset provides 100,000 images as a training set and 99,996 images as a validation set for the conventional scenario, and part of the images of the training set and validation set are shown in Fig. 6A. In order to evaluate the recognition ability of the model in various extreme situations, the experiments use four important test sets of extreme situations in CCPD, which are Challenge, Blur, DB, and FN. Challenge (50,003 images) is used to test the performance of the model in the case of challenging license plate images; Blur (20,611 images) is used to test the performance of the model in the case of blurred license plate images; DB (10,132 images) is used to test the model’s performance in the case of too low or too high illumination; and FN (20,967 images) is used to test the model’s performance in the case of low resolution. Some of the images of the four test sets are shown in Fig. 6B. In addition, the CCPD dataset also provides a comprehensive test set, which is a comprehensive sampling of several extreme cases, and the images in this part are corrected by affine transformation.

Figure 6 Example of CCPD dataset.

(A) Train set and Validation set; (B) Challenge set, Blur set, DB set, FN set; (C) Rotate set, Tilt set, No Affine Transformation set.

In addition, in order to evaluate the spatial adaptation ability of the model, the experiments were evaluated using the Rotate and Tilt test sets in CCPD without affine transformation. As shown in Fig. 6C, rotate and Tilt test sets in CCPD with the test set without affine transformation. As shown in Fig. 6C, Rotate (10,053 images) is used to test the performance of the license plate in the case of rotation; the license plate in Tilt (30,216 images) is used to test the performance of the license plate in the case of extreme angles; and finally, the experiment is evaluated on the test set without affine transformation correction, which is used to evaluate the performance of the model without the use of corrective transformations.

Training parameter settings

The experiments were conducted using the processed CCPD training set with the training parameters shown in Table 1.

Table 1 Neural network training parameters.

Parameter	Value	
Epoch	10	
Image size	(94, 24)	
Batch size	256	
Learning rate	1.00E−03	
Optimizer	Adam	
Weight decay	2.00E−05	
Dropout rate	0.5	

Assessment of indicators

The evaluation of the license plate recognition task should not only refer to whether the model is correct in recognizing characters, but also whether the model is correct in recognizing the number of characters. Therefore, TN1 is defined as the number of samples in which the number of characters is recognized incorrectly, TN2 is defined as the number of samples in which the characters are recognized incorrectly, and TP is defined as the number of samples in which the number of characters is recognized correctly and the characters are recognized correctly. The evaluation indexes are introduced: Length Consistency Rate (LCR) and Accuracy. The calculation formula is shown in Eqs. (7) and (8):

(7) LCR=TN2+TPTN1+TN2+TP

(8) Accuracy=TPTN1+TN2+TP.

The indicator LCR can reflect the accuracy of the number of characters recognized; the closer to 1 means that the number of characters recognized accurately, the greater the number of characters; the indicator Accuracy can reflect the number of characters and the accuracy of character recognition at the same time.

Experimental results and visualization

The LPRNet method based on graphically supervised signal-assisted training proposed in this article is used to do side-by-side comparison experiments with models such as AlexNet (Krizhevsky, Sutskever & Hinton, 2017), ResNet (He et al., 2016), LPRNet, CRNN, EfficientNet (Tan & Le, 2019) and the experimental parameters are shown in Table 1. The accuracy of each model is evaluated on the validation set with different test sets, and the results are shown in Table 2. The code related to the experiments in this article is publicly available and can be accessed via the following GitHub link: https://github.com/GENERjia/License-Plate-Recognition-Based-on-Graphically-Supervised-Signal-Assisted-Training. Or download it via the DOI link: https://doi.org/10.5281/zenodo.15647076.

Table 2 Comparison of the accuracy of scene license plate recognition models on each CCPD test set.

Bold entries indicate the best performance in each rows.

DataSet	Metrics	CRNN	LPRNet	AlexNet	ResNet-18	EfficientNet b1	Ours (Compared to LPRNet)	
val	LCR	0.9981	0.9992	0.9954	0.9959	0.9926	0.9996 (+0.0004)	
Accuracy	0.9903	0.9964	0.9861	0.9845	0.9720	0.9972 (+0.0008)	
test	LCR	0.7212	0.7278	0.6503	0.8629	0.872	0.8538 (+0.1259)	
Accuracy	0.4715	0.5739	0.4508	0.5523	0.3938	0.6370 (+0.0631)	
blur	LCR	0.6446	0.5219	0.4365	0.8473	0.8495	0.7954 (+0.2735)	
Accuracy	0.3386	0.3468	0.2626	0.4580	0.3648	0.4558 (+0.1090)	
challenge	LCR	0.7640	0.7089	0.6697	0.8880	0.9122	0.8461 (+0.1371)	
Accuracy	0.4921	0.5386	0.4602	0.5865	0.4570	0.6022 (+0.0637)	
db	LCR	0.5764	0.5473	0.4252	0.7700	0.6115	0.7769 (+0.2296)	
Accuracy	0.3291	0.4053	0.2782	0.3861	0.1603	0.5144 (+0.1091)	
fn	LCR	0.7548	0.7755	0.7199	0.8609	0.8885	0.8681 (+0.0927)	
Accuracy	0.5320	0.6461	0.5166	0.5904	0.4417	0.6963 (+0.0501)	
rotate	LCR	0.7835	0.8861	0.8242	0.9121	0.9170	0.9190 (+0.0329)	
Accuracy	0.6419	0.8236	0.6792	0.7142	0.4805	0.8413 (+0.0177)	
tilt	LCR	0.7067	0.8081	0.7329	0.8482	0.8826	0.8939 (+0.0857)	
Accuracy	0.4768	0.6916	0.4995	0.5351	0.3256	0.7470 (+0.0554)	
Average	LCR	0.7437	0.7468	0.6818	0.8732	0.8658	0.8691 (+0.1222)	
Accuracy	0.5340	0.6278	0.5166	0.6009	0.4494	0.6864 (+0.0586)	

In order to compare the performance of different models, the computational volume, number of parameters, FPS, and average accuracy of different models are counted, as shown in Table 3.

Table 3 Comparison of FLOPs, number of parameters, FPS on CPU and average accuracy among models.

Models	FLOPs	Params	FPS	Accuracy	
CRNN	0.95 G	7.3 M	92.30	0.5340	
LPRNet	0.29 G	0.4 M	93.24	0.6278	
AlexNet	0.63 G	3.8 M	90.03	0.5166	
ResNet-18	1.09 G	11.2 M	76.39	0.6009	
EfficientNet b1	0.32 G	9.2 M	83.32	0.4494	
Ours	0.29 G	0.4 M	93.24	0.6864	

From Table 3, it can be seen that the proposed algorithm achieves the highest accuracy with the lowest number of parameters and computations. It proves the effectiveness of the proposed algorithm. The computational amount, number of parameters, and average accuracy of each model in Table 3 are visualized as shown in Fig. 7.

Figure 7 Visualization of models comparison.

The comprehensive comparison of proposed algorithm with other common license plate recognition algorithms is shown in Fig. 8. Figure 8A shows the comparison of the computational volume and accuracy between this article and several common models; the x-axis indicates the computational volume of the model; the size of the diameter of the dots indicates the number of model parameters; and the y-axis indicates the average accuracy in the test set. Figure 8B shows the comparison of the accuracy of several common models on different test sets.

Figure 8 Comparison of this article’s algorithm with other common algorithms.

(A) Model accuracy vs. computational cost; (b) comparison of models on different datasets.

The experiments were recorded for the metrics during the training of CRNN, LPRNet, ResNet, and AlexNet, as shown in Fig. 9.

Figure 9 Comparison of losses and accuracy in the validation set of the training process model.

The results show that LPRNet is much higher than other models in both accuracy and convergence speed.

The experiment uses LPRNet as a baseline and introduces graphically supervised signals on the basis of LPRNet. The experiment records the metrics during the training process of the LPRNet model and proposed algorithm, as shown in Fig. 10.

Figure 10 Comparison of accuracy and loss between LPRNet and LPRNet validation set based on graphically supervised signal-assisted training during the training process.

The results show that the accuracy of proposed algorithm is slightly higher than that of the original model on the validation set, and higher accuracies are obtained on all test sets, which proves that proposed algorithm can show good recognition results. In terms of the convergence speed and loss of the model, proposed algorithm is similar to the original model, i.e., this article’s algorithm does not need more iterations for optimization while improving the accuracy.

In order to more intuitively reflect the improved effect of the algorithm, different license plates from the experimental findings will be shown in the experiment. The results are shown in Fig. 11.

Figure 11 Recognition effect of different models on license plate in different cases.

For low light, blur, low resolution, angular offset, motion offset, etc., proposed algorithm can get more and more accurate detection results.

In order to further analyze the results, the model output can be visualized, as shown in Fig. 12A. It is the classification output of LPRNet, and proposed algorithm is more accurate for the identification of blank characters and can distinguish between characters and the interval between them; Fig. 12B is the prediction output of proposed algorithm for the image label. The prediction output of the image label can basically realize the extraction of effective information from the image, and the prediction of the image label can basically react to the classification output situation.

Figure 12 Model output results.

(A) Classification output; (B) prediction output of proposed algorithm for the image label.

In order to study the recognition of each character in the dataset, we conducted ablation experiments to better validate the effectiveness of proposed algorithm. And because of the uneven balance of the province character categories in the dataset, the experiments were only conducted on the recognition of numeric characters and alphabetic characters, as shown in Table 4.

Table 4 Recognition of numeric and alphabetic characters.

	LPRNet	Ours	
Char	Precision	Recall	F1-score	Precision	Recall	F1-score	
0	0.814	0.743	0.777	0.87	0.819	0.844	
1	0.795	0.771	0.783	0.837	0.87	0.853	
2	0.817	0.782	0.799	0.871	0.886	0.879	
3	0.858	0.779	0.817	0.913	0.852	0.882	
4	0.796	0.775	0.785	0.82	0.862	0.841	
5	0.803	0.784	0.794	0.877	0.87	0.874	
6	0.855	0.766	0.808	0.903	0.858	0.88	
7	0.828	0.768	0.797	0.894	0.86	0.877	
8	0.839	0.765	0.8	0.893	0.832	0.861	
9	0.819	0.797	0.808	0.902	0.874	0.888	
A	0.947	0.955	0.951	0.92	0.971	0.945	
B	0.919	0.679	0.781	0.904	0.74	0.814	
C	0.893	0.795	0.841	0.875	0.875	0.875	
D	0.9	0.639	0.747	0.855	0.72	0.781	
E	0.889	0.771	0.826	0.907	0.817	0.86	
F	0.871	0.763	0.813	0.876	0.858	0.867	
G	0.863	0.743	0.799	0.905	0.78	0.838	
H	0.926	0.701	0.798	0.932	0.773	0.845	
J	0.87	0.782	0.824	0.852	0.873	0.862	
K	0.915	0.733	0.814	0.909	0.822	0.863	
L	0.926	0.686	0.788	0.907	0.814	0.858	
M	0.905	0.764	0.828	0.904	0.847	0.874	
N	0.89	0.717	0.794	0.897	0.801	0.847	
P	0.904	0.775	0.835	0.935	0.833	0.881	
Q	0.808	0.814	0.811	0.811	0.831	0.821	
R	0.909	0.757	0.826	0.944	0.8	0.866	
S	0.801	0.78	0.791	0.884	0.806	0.843	
T	0.875	0.711	0.785	0.895	0.747	0.814	
U	0.851	0.708	0.773	0.831	0.795	0.813	
V	0.813	0.768	0.79	0.849	0.811	0.83	
W	0.718	0.801	0.757	0.786	0.86	0.822	
X	0.869	0.782	0.823	0.863	0.861	0.862	
Y	0.858	0.791	0.823	0.842	0.856	0.849	
Z	0.889	0.731	0.803	0.901	0.764	0.826	
Average	0.860	0.761	0.806	0.881	0.830	0.854	

Visualize the increase of precision, recall, and F1 score indicators for each character using the proposed algorithm in the form of a bar chart, and analyze the increase using a histogram, as shown in Fig. 13. For the recognition of numeric characters, proposed algorithm not only enhances the precision rate of character recognition but also improves the recall rate, enabling it to recognize the characters that the original algorithm can not. As for the alphabetic characters, while proposed algorithm reduces the precision rate of the recognition of some characters, it can greatly improve the recall rate of character recognition so that the F1 score can be improved.

Figure 13 Increase in precision, recall, and F1 score metrics for each character.

For LPRNet and proposed algorithm, experiments were performed to visualize Grad-CAM (Selvaraju et al., 2017) on both models as shown in Fig. 14 in order to observe the regions of interest on the pictures for each category.

Figure 14 Comparison of Grad-CAM visualization for different models.

By observing the visualization results, it can be found that proposed algorithm can pay better attention to the correct position of each character without focusing too much on the other regions, which is especially obvious in the model’s understanding of the “-” character, and proposed algorithm can better capture the segmentation and positioning relationship between characters.

In order to evaluate the spatial adaptation ability of the model, the experiments are conducted to evaluate LPRNet and proposed algorithm in the Rotate, Tilt test set, and the test set without affine transformation respectively, and the results are shown in Table 5.

Table 5 Accuracy of the model when the image undergoes a change in spatial angle.

DataSet (no affine)	LPRNet	Ours	Increase	
Rotate	0.0017	0.0006	−0.11%	
Tilt	0.0020	0.0010	−0.10%	
Test	0.1946	0.2110	1.64%	

The results show that in the case of large angle change, both LPRNet and proposed algorithm can not achieve better results, and the effect of proposed algorithm is slightly lower than that of LPRNet. While in the case of no affine transformation correction, i.e., test set, proposed algorithm is slightly higher than that of LPRNet, which indicates that this article’s algorithm still achieves better results in the case of small angle change.

Discussion

Our study demonstrates the effectiveness of employing graphically supervised signals (GSS), derived from text labels, to assist the training of a lightweight LPR model, LPRNet. Key findings on the CCPD dataset reveal that this auxiliary training significantly improves recognition accuracy across its various challenging subsets, including those with blur, low illumination, and varying resolutions, without imposing any additional computational burden during inference. The overall average accuracy on the CCPD test sets improved by 5.86%, with notable gains in recall (average of 6.9%), suggesting the model learns to recognize characters it previously missed, thereby enhancing its practical utility.

The mechanism behind this performance enhancement, particularly when compared to the baseline LPRNet, likely lies in the richer and more discriminative feature representations learned by the shared backbone network. By requiring the backbone to simultaneously optimize for both sequence recognition (via CTC loss) and graphical reconstruction of the GSS (via MSE loss on the image label), the model is implicitly encouraged to learn features that capture not only the semantic identity but also the crucial visual and spatial structure of characters more effectively. This dual supervision appears to lead to more robust feature extraction, which is supported by the Grad-CAM visualizations (Fig. 14) indicating more focused attention on relevant character regions with our method. This “smarter training strategy” allows us to boost performance while preserving the inherent lightweight nature of the LPRNet architecture.

When positioning our approach against other LPR methods, especially those that might report higher absolute accuracy by employing larger backbones, complex attention mechanisms, or extensive post-processing, our method offers a distinct practical advantage. It provides a significant accuracy uplift for an already efficient model, making it particularly suitable for deployment on resource-constrained embedded devices where the trade-off between accuracy and computational cost (Params, FLOPs, inference speed) is critical. While some state-of-the-art models achieve superior accuracy, they often do so at the expense of substantially higher computational demands, rendering them less feasible for real-time, on-device LPR. Our GSS-assisted LPRNet, therefore, presents a compelling balance, outperforming its baseline significantly and offering competitive performance within the lightweight model category.

Despite these promising results, our approach has limitations. While the inference cost remains unchanged, the introduction of the GSS and the auxiliary training branch does lead to an increase in training time and memory footprint during the training phase. Specifically, training our model required approximately 3 h, representing a notable increase compared to the 0.75 h needed for the baseline LPRNet under identical experimental conditions. A more subtle limitation arises from the GSS generation process. The current static alignment method, based on average dataset statistics, may not perfectly align the generated image label with the specific geometry of every individual input license plate. This potential misalignment could contribute to the observed slight decrease in accuracy on test sets with extreme spatial angles (e.g., CCPD-Rotate, as shown in Table 5), where precise spatial correspondence is paramount. Furthermore, the model’s performance might be sensitive to unconventional fonts or highly diverse plate layouts not well-represented in the CCPD dataset.

To address these limitations and further improve the method, future work could explore several avenues. Developing dynamic alignment techniques to generate GSS that adapt more closely to each input sample’s unique perspective could mitigate issues with extreme angles. Investigating the impact of using multiple fonts or a more diverse font generation strategy for GSS could enhance robustness to varied real-world plate appearances. Additionally, exploring alternative auxiliary tasks or loss functions beyond simple MSE reconstruction might yield further benefits. Finally, evaluating the proposed GSS-assisted training on other lightweight LPR architectures and diverse international LPR datasets would be crucial for assessing its broader applicability and generalizability.

Conclusion

This article introduced an effective license plate recognition algorithm enhanced by graphically supervised signal-assisted training. Our study demonstrates the robustness of this approach. A key advantage is the significant improvement in model performance achieved without altering the neural network’s inference structure or increasing its computational cost.

Experimental results show substantial gains. Average accuracy in complex scenarios increased by 5.86%. The length consistency rate improved by 12%. Furthermore, average character precision rose by 2.1%, and average recall increased by 6.9%. These improvements indicate our method effectively enhances character recognition, enabling the model to identify characters previously unrecognized. The primary cost for this enhanced performance lies in the need to generate graphical supervisory signals from training data, which also moderately increases training time and memory usage during the training phase.

The proposed training process is broadly applicable to many convolutional neural networks. Future research could extend this GSS-assisted training to other image recognition tasks. This includes potential applications in non-textual image classification. Exploring methods to effectively mine graphical supervisory signals from regular, non-textual images will be a key challenge and an exciting direction for future work.

Supplemental Information

Supplemental Information 1 English translations of the Chinese Provinces.

Supplemental Information 2 Code.

Supplemental Information 3 Translations for non-English in code.

Additional Information and Declarations

Competing Interests

The authors declare that they have no competing interests.

Author Contributions

Dianwei Chi conceived and designed the experiments, performed the computation work, prepared figures and/or tables, authored or reviewed drafts of the article, and approved the final draft.

Zehao Jia performed the experiments, performed the computation work, prepared figures and/or tables, and approved the final draft.

Lizhen Liu analyzed the data, prepared figures and/or tables, and approved the final draft.

Data Availability

The following information was supplied regarding data availability:

The data is available at GitHub and Zenodo:

- https://github.com/detectRecog/CCPD.

- Xu, Z., & Jia, Z. (2025). CCPD (Chinese City Parking Dataset) for “Research on license plate recognition based on graphically supervised signal-assisted training” [Data set]. Zenodo. https://doi.org/10.5281/zenodo.15647076.

The code is available in the Supplemental File and at GitHub:

https://github.com/GENERjia/License-Plate-Recognition-Based-on-Graphically-Supervised-Signal-Assisted-Training.

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
