# Peer review of "Research on license plate recognition based on graphically supervised signal-assisted training"

_PeerJ Computer Science, doi:10.7717/peerj-cs.2989_

## Round 0.1 · original submission · Major Revisions

Five reviewers have commented on yuor manuscript.

**Language Note:** The review process has identified that the English language must be improved. PeerJ can provide language editing services - please contact us at [email protected] for pricing (be sure to provide your manuscript number and title). Alternatively, you should make your own arrangements to improve the language quality and provide details in your response letter. – PeerJ Staff

Reviewer 1 ·

Basic reporting

In this manuscript, authors propose a license plate recognition algorithm, which uses LPRNet (License Plate Recognition Network) as the base model and incorporates graphically supervised signals for assisted training, aiming to improve the accuracy of the model by improving its training process, and to realize a lightweight and highly accurate license plate recognition model. This study and design are interesting. A well-known CCPD dataset is used in experiments. This dataset has huge diversity and is a challenging dataset.

I propose that the Related Works section should be a bit more improved with the latest references. Also more highlight your contributions. Currently, your contributions are just listed as 1, 2, 3,... Make them bullets and explain them to readers.

Experimental design

The Methods section seems to be messed up with several equations. It makes it hard for the readers to understand and keep the flow. I propose that, if authors could include a pseudocode of their proposed method, it would be great for readers to understand the actual mechanism of the proposed method.

Also, in the Methods section, mention the parameters below each equation.

Validity of the findings

The Experiment and Design section is good. It lists the parameters being used and the dataset description. It also lists the availability of code on GitHub. I have several suggestions for this section.

1. Please put a dedicated Discussion section here and show your findings and observations on the CCPD dataset.

2. In this section, the computational complexity of the proposed method and the compared methods is missing. I propose to include the training/test times of the proposed method and the compared methods.

3. Also, put the limitations of your proposed method. In what environment can it be improved?

4. It is not discussed why and how the proposed method outperforms the several compared methods.

Additional comments

This manuscript can be improved further. Please put the pseudo code of the developed method, show contributions clearly, and discuss your method in detail.

Reviewer 2 ·

Basic reporting

-

Experimental design

The comparison methods used in Table 2 of the experiment are too outdated. The author should compare more algorithms from the past two years. Although the author's accuracy is lower than some existing algorithms, the algorithm's speed is relatively fast, which still makes it valuable for applications. There is no need to deliberately conceal the accuracy of better methods.

Validity of the findings

-

Reviewer 3 ·

Basic reporting

-

Experimental design

-

Validity of the findings

The authors use the CCPD dataset, which has been widely utilized in prior studies. A comparative analysis of results with other works using the same dataset is necessary for evaluating the proposed method’s relative performance.

Additional comments

Related Works Section: Typically, this section focuses on prior research on the same topic. Presenting foundational concepts here is unusual. The section currently lacks citations, which should be included to support the discussion.

Tables 2–6 reference sources using bracketed citations, but the corresponding references are missing. The reference list is formatted alphabetically without numbering, indicating a potential last-minute change in the manuscript template. This should be corrected.

Some equations have formatting issues.

·

Basic reporting

The English language is not good, for example, both abstract part lines (15 till 26) and introduction parts lines (119-124, 139-145, 180-186), rewrite and edit sentences as short sentences to be clearer.

Experimental design

There is no clear contribution to this paper. The techniques used already exist.

Validity of the findings

Rewrite and edit the conclusion sentences as short sentences to be clearer.

Additional comments

1- Please indicate clearly where the new contribution to your paper is. The techniques used already exist.
2- In both the abstract part lines (15 till 26) and introduction part lines (119-124, 139-145, 180-186), rewrite and edit sentences as short sentences to be clearer.
3- The authors mentioned in line (372) that there are four important test sets of extreme situations in CCPD, which are Challenge, Blur, DB, and FN. The authors should mention how many samples are in each set.
4- The authors did not mention character recognition in this work.
5- The limitations of your proposed method are absent and not found in your paper.
6- The authors did not mention the speed of processing in this work.
7- In Table 5, the accuracy of the proposed model in spatial angle decreased when compared with LPRNet. What is your interpretation of this situation?
8- In page (9), 4.7. In the comparison experiment, the second paragraph line (5), the author mentioned Table 5, but there is no Table 5 in the paper.
*** Strength points: Please write down the strong points of the manuscript
No comment
*** Weakness Points: Please write down the weak points of the manuscript
• There is no clear contribution to this work. Please mention your contribution.

Reviewer 5 ·

Basic reporting

Lines 179-207: "Please provide the reference for LPRNet in lines 179-207. Explain why this model is utilized for license plate recognition. Are there any studies that have previously employed this model? If so, are they specifically related to plate recognition or general object recognition? Please elaborate further in the revised manuscript."

Line 210: "Please clarify what you mean by 'Graphically supervised signals.' Use terminology that is commonly understood in this field and include references to the origins of this method."

Figure 1: "Please provide more detail about Figure 1. What do the abbreviations in the figure mean, and how does it function?"

Experimental design

Table 3: "According to the results for 'Our model,' the parameters (params), floating point operations (FLOPs), and frames per second (FPS) are all equivalent to LPRNet; however, the accuracy differs. Could you explain why this discrepancy exists? Why does the accuracy vary while all other metrics are the same as LPRNet? Please elaborate on the reasons for this difference and highlight the distinctions between your model and LPRNet."

Real-World Testing: "Is it possible to test your model in a real-world scenario, such as using a real camera to assess whether the model can effectively recognize license plates in live environments? How quickly can the model detect a plate? What is the optimal distance for the camera from the plate? This information would be beneficial for readers and would provide insights for decision-makers on how to implement this technology in the real world."

Validity of the findings

This section addresses the reliability of the findings and their interpretation in practical scenarios. Suggestions for testing the model under real-world conditions and addressing discrepancies in results, such as those seen in Table 3 (accuracy compared to other metrics), are crucial for enhancing the credibility and applicability of the findings.

---

## Round 0.2 · accepted · Accept

The authors have addressed all concerns.

Reviewer 3 ·

Basic reporting

No comment

Experimental design

No comment

Validity of the findings

No comment

Additional comments

The authors still have not provided a direct comparison with existing methods, as requested by me and other reviewers in the previous round. Their justification that the use of an affine transformation prior to network input makes direct comparison invalid is unconvincing. Since the affine transformation is a part of their approach, a comparison of the final outcomes remains necessary. That said, given the overall contribution of the paper, particularly in making the process more lightweight, I am recommending acceptance, while noting the above reservations.

·

Basic reporting

The paper is well organized. The literature review is extensive and up-to-date, citing relevant works.

Experimental design

The author revised the design according to reviewer's requirements.

Validity of the findings

All underlying data have been provided; they are robust, statistically sound, & controlled.

Reviewer 5 ·

Basic reporting

The authors have addressed all the revision issues well.

Experimental design

The authors have addressed all the revision issues well.

Validity of the findings

The authors have addressed all the revision issues well.